# Modeling and Design Parameter Optimization to Improve the Sensitivity of a Bimorph Polysilicon-Based MEMS Sensor for Helium Detection

**DOI:** 10.3390/s24113626

**Published:** 2024-06-04

**Authors:** Sulaiman Mohaidat, Fadi Alsaleem

**Affiliations:** 1Department of Mechanical and Materials Engineering, University of Nebraska-Lincoln, Lincoln, NE 68588, USA; smohaidat2@huskers.unl.edu; 2Department of Architectural Engineering, Durham School of Architectural Engineering and Construction, University of Nebraska-Lincoln, Omaha, NE 68182, USA

**Keywords:** MEMS, helium sensing, bimorph, COMSOL

## Abstract

Helium is integral in several industries, including nuclear waste management and semiconductors. Thus, developing a sensing method for detecting helium is essential to ensure the proper operation of such facilities. Several approaches can be used for helium detection, including based on the high thermal conductivity of helium, which is several times higher than air. This work utilizes the high thermal conductivity of helium to design and analyze a bimorph MEMS sensor for helium sensing applications. COMSOL Multiphysics software (version 6.2) is used to carry out this investigation. The sensor is constructed from poly-silicon and SiO_2_ materials with a trenched cantilever beam configuration. The sensor is electrically heated, and its morphed displacement depends on the surrounding gas’s composition, which decreases in the presence of helium. Several factors were investigated to probe their effect on the sensor’s sensitivity to helium, including the thickness of the poly-silicon layer, the configuration of the trench, and the thickness and location of SiO_2_ layer. The simulations showed that the best performance, up to 2 ppm helium detection level, can be achieved with thinner beams and medium trench lengths.

## 1. Introduction

Dry casks are used to store spent nuclear fuel in an inert and dry environment in which fuel is cooled by natural convection. Helium pressurizes metal canisters within the storage system to transfer heat from spent fuels through natural convection inside the canister. Helium leakage is an abnormal operating condition for dry casks [1]. Microelectromechanical systems (MEMS) are microscale devices fabricated using semiconductor technology. They integrate mechanical and electrical components for sensing and actuation. MEMS devices can interact with several energy domains to gain information about an environment, including electrical, thermal, mechanical, chemical, magnetic, and radiation. MEMS can utilize input energy for mechanical motion through actuation mechanisms such as electrostatic, electrothermal, electromagnetic, and piezoelectric actuation [2].

MEMS provides small-size, low-cost, and low-power gas sensing platforms. Several mechanisms are employed through MEMS technology to detect gases, such as acoustic, optical, thermal, and electrochemical technology [3,4]. Thermal conductivity gas sensing depends on the thermal conductivity of the target gas. The target gas can affect the heat transfer from a high-temperature structure to a low-temperature region or device. The heat transfer from a heated structure to the surrounding gas will be higher for gases with a higher thermal conductivity. Thermal conductivity-based gas sensors have advantages, including eliminating catalysts and adsorbents [5,6]. Moreover, thermal conductivity gas sensors have been found in many applications, including coal mines, natural gas monitoring (for methane sensing), and light gases such as hydrogen [7].

This work uses COMSOL Multiphysics finite element software (version 6.2) to model helium detection in the air using an electrothermal bimorph MEMS structure. A bimorph cantilever MEMS device is created from a poly-S/SiO_2_ structure. The sensing principle utilizes the difference in thermal conductivity between air and helium. Any heated microstructure that is surrounded by helium gas will have a lower temperature when compared to air due to the higher thermal conductivity of helium (0.152 W/m · K at 300 K) compared to that of air (0.0263 W/m · K at 300 K) [8]. The bimorph devices comprise two or more layers. Each layer will have its thermal expansion coefficient, resulting in the device bending when heated. The displacement will depend on the device’s temperature, which, in turn, depends on the surrounding gas’s thermal properties [9].

The finite element modeling used in this work is an effective tool for the design and analysis of engineering structures, including strain sensors [10] and a polysilicon-based temperature pressure sensor [11]. It was also coupled with the finite volume method for the analysis and design of electrothermal MEMS actuators [12]. The Multiphysics modeling approach used in this work couples and incorporates several physical systems (modeled through PDEs) into a single model [13,14,15,16,17,18].

## 2. Materials and Methods

The structure was made from a poly-silicon cantilever device with a SiO_2_ layer that can be placed below or above the Si layer. The length of the device was 300 µm. The device comprised two segments, each 150 µm in width, connected by a trench structure with a width of 20 µm, as shown in Figure 1. The formed trench ensured uniform electrical current distribution. The cantilever had 400 µm × 400 µm pads that are fixed.

A gap separates the cantilever from the substrate, forming a capacitive readout approach for the modeled sensor. The surrounding gas was modeled as a surrounding domain, with a total domain size of 1020 µm × 900 µm × 200 µm. The Si material was modeled as a poly-silicon with temperature-dependent thermal conductivity and thermal expansion coefficients. The poly-silicon resistivity was assumed to be 0.03 Ω cm, which is within the reported resistivity values of poly-silicon [19].

The device used joule heating. The temperature-dependent displacement due to heating was modeled using the thermal expansion interface, which combined heat transfer and solid mechanics. The external boundaries were maintained at room temperature of 293.15 K.

The heat transfer mechanism from the heated sensor to the ambient environment was modeled as conduction, not convection, because of the microscale nature of the design (i.e., the length was 300 µm) [20]. Mathematically, the heat transfer mechanism could be determined by calculating the Rayleigh number, according to the following
(1)$$R_{a}=\frac{g\;\beta \;\left(T_{s}-T_{\infty }\right)L^{3}}{\;v\;\alpha }$$
where *R_a_* is the Rayleigh number, *g* is the acceleration of gravity, *ν* is the kinematic viscosity (air = 45.1056 × 10^−6^ m^2^/s and helium = 339.37 × 10^−6^ m^2^/s), *α* is the thermal diffusivity (air = 45.1056 × 10^−6^ m^2^/s and helium = 339.37 × 10^−6^ m^2^/s), *β* is the volumetric thermal expansion coefficient (1/*T*), *T_s_* is the surface temperature, $T_{\infty }$ is the ambient temperature, and *L* is the length. All of the properties were obtained at the film temperature: *T_f_*= *(T_s_ +*
$T_{\infty })$/2. Assuming the surface temperature was 800 K, and the ambient temperature was 293.15 K and the film temperature was 546.575 K.

Using Equation (1), the Rayleigh number for air was 0.0825069, and for helium it was 0.00141451. Thus, the flow nature of the intended application did not involve forced flow as the value of the Rayleigh number was very small (<10^3^), and it could be assumed that the heat transfer mechanism was conduction in the gas [20]. The heat conduction equation through the gas can be described by the following:*q* = −*k A* ∗ *T*(2)
where *q* is the heat transfer rate, *k* is the material’s thermal conductivity, *A* is the normalized cross-sectional area to thickness, and ∇*T* is the temperature difference.

The MEMS device was subjected to DC heating voltage to obtain a maximum temperature of around 800 K in the air condition so as to minimize the creep effect, which was reported to start around 700 °C (973 K) [21].

Several variations of the poly-Si layer thickness, gap thickness, SiO_2_ layer location, and trench configuration were tested computationally to improve the sensor sensitivity. They will be presented in the Section 3. The sensor’s sensitivity was related to the difference in the displacement of the cantilever tip between air and helium situations (a higher deflection difference means a more sensitive response to the presence of helium). The cantilever displacement can be measured using a non-contact method such as a laser beam or a capacitive readout circuitry.

## 3. Results

### 3.1. Basic Design

The basic model was made from a ploy-Si layer with a 20 µm thickness, with a 500 nm SiO_2_ layer below it and a gap thickness of 500 nm. The device reached a maximum temperature of 800 K in the surrounding air when subjected to 9.896 V DC voltage. At the same voltage, the maximum temperature dropped to around 523 K in the presence of helium. The corresponding displacement at the air condition was 0.0547 µm, and for helium it was 0.020 µm. The displacement of the sensor in the *z*-direction for both cases is shown in Figure 2A,B.

### 3.2. Geometry Effect

The effect of the beam thickness and the gap below the MEMS sensor on its performance was studied next. The gap was varied (0.5, 1, 2, 3, 4, and 5 µm) and the beam thickness was varied (5, 10, and 20 µm). The sensitivity for all these combinations is illustrated in Figure 3. Also, Table 1 shows the heating voltage required to achieve the baseline maximum temperature of 800 K before applying helium.

It is clear from Figure 3 that the 5 µm beam had the highest sensitivity, which can be explained by its reduced stiffness compared with the 10 µm and 20 µm beams, as seen in Figure 4. The stiffness increased to around 57 times from 5 µm to 20 µm thickness. A stiffer structure will show more resistance to thermal expansion, even if it has the same temperature as another device with a lower stiffness.

Another behavior noticed in Figure 3 is the increase in sensitivity as the gap distance increased to a specific limit, especially for the thinner beams. The increase in the sensitivity could be attributed to the increase in temperature of the beams due to the increases in the thermal resistance of the gas layer [22], and, hence, the reduction in the heat flux between the beam and the gas, as shown in Figure 5. As the model accounted for thermal conductivity and thermal expansion coefficients’ dependency on temperature, this temperature resulted in higher thermal gradients, leading to higher displacement differences between air and helium.

Finally, Table 1 shows that thinner beams required higher input voltages to operate. This was attributed to the reduction of the beam electrical resistance with increased beam thickness [23], as shown in Figure 6.

### 3.3. Effect of the Trench

In the basic design, the ploy-silicon trench connecting the two cantilever segments started at 280 µm, as shown in Figure 1. This section studied the effect of the starting of the connection structure. The starting ranged from 80 µm to 280 µm with 20 µm increments. The thickness of the poly-silicon was maintained kept at 5 µm, and the gap gas thickness was 5 µm—the best geometry parameters from the previous section. The SiO_2_ layer was 500 nm thick and situated below the silicon layer.

The sensitivity of the sensor as a function of the trench start position is shown in Figure 7. The figure shows the nonlinear effect of the trench configuration on the sensor’s performance. At the beginning, the sensitivity increased with the starting position of the silicon to a maximum at 160 µm positions, then decreased to a minimum at the highest value of the starting position of the silicon.

The effect of the trench configuration on the electrical resistance of the MEMS sensor is shown in Figure 8. The electrical resistance increase based on the starting position of silicon in the trench could be explained by the change in the cross-sectional area of the MEMS sensor. For example, with the trench starting at 80 µm, the sensor had the largest cross-sectional area and thus the lowest resistance. This low electrical resistance, as shown in Figure 9, produced the highest heat flux in air or helium conditions. With the increase in the starting position, the area was reduced, leading to the highest resistance and lowest heat flux at the 280 µm starting position, corresponding to the base design. However, while the heat flux decreased as the start position increased, the heat flux difference (the maximum difference between air and helium) occurred around the middle. This nonlinear behavior was also observed in the input voltage values required to heat the sensor around 800 K in an air condition, as shown in Table 2.

### 3.4. Effect of the SiO_2_ Layer Thickness and Location

Next, we investigated the effect of the SiO_2_ layer on the sensor sensitivity. Figure 10 shows the sensor sensitivity as a function of the layer thickness. The figure shows a linear relationship.

A final investigation was conducted by placing the SiO_2_ layer above the cantilever beam. While most of the above analyses and findings were still applied when the SiO_2_ layer was placed underneath the cantilever, a significant difference is that the beam would move upward. Thus, its deflection had no physical limit like when it was moved down and was limited by the substrate gap. This up-free movement allowed for the simulation of a less-stiff thinner beam of 2 µm in thickness compared with the beams explored previously with thicknesses of 5 µm, 10 µm, and 20 µm.

The trench effect was repeated for this configuration with the same variation in starting position (from 80 µm to 280 µm with 20 µm increments). The sensitivity based on this variation is shown in Figure 11. Figure 11 shows a similar sensitivity trend to that shown in Figure 7, but with an improvement of four times the sensitivity at the maximum trench starting point. Again, this significant improvement was due to placing the SiO_2_ layer on top of the beam, allowing a thin beam to move up freely. The increase in sensor sensitivity due to placing the SiO_2_ layer on top of the beam, as shown in Table 3, allowed for detecting a fraction of helium with a reasonable sensor deflection difference.

## 4. Discussion

The above results showed the utilization of finite element modeling (COMSOL Multiphysics) to design a thermal conductivity gas sensor to detect helium in air. The sensing approach was based on an electrothermal bimorph MEMS constructed from Si/SiO_2_ materials. The bimorph sensor was designed as a cantilever beam from poly-silicon material with a constant resistivity of 0.03 Ω cm. The poly-silicon had a trench to help distribute the temperature across the beam equally. The second layer of the sensor was a 500 nm SiO_2_ layer. The thermal conductivity and thermal expansion coefficients were assumed to be temperature-dependent properties. The displacement of the MEMS sensor was dependent on the temperature, which was dependent on the surrounding gas, so the displacement of the cantilever beam’s tip depended on the composition of the surrounding gas with reduced displacement with an increased presence of helium in the surrounding gas.

Several sensor configurations based on the poly-silicon layer thickness, dimensional variations, and the location of the SiO_2_ layer (below or above the poly-silicon layer) were explored computationally. The simulation was conducted with a maximum temperature limit of 800 K to minimize the effect of creep. The sensitivity was based on the difference in the displacement between air and helium.

It was found that thickness was an important design parameter, with thinner beams having a better performance due to their low stiffness. However, the resistance increase with a decreased beam thickness led to an increase in the input voltage to achieve the same required temperature (800 K), as seen in Table 1. The gap between the sensor and the trench configuration also affected the sensor’s sensitivity. The sensitivity of the 5 µm thickness MEMs sensor increased by more than two times when the gap thickness increased from 0.5 µm to 5.0 µm. The 5 µm thickness with the 5.0 µm gap had more than 12 times more displacement compared with a 20 µm thickness sensor with the same gap dimensions.

The effect of the trench configurations showed a nonlinear response with an optimal point around the trench length, which was almost the base design’s length. The explanation of this complex behavior was not trivial and was shown to correlate with the heat flux difference between air and helium, as shown in Figure 9.

A Figure of Merit (*FoM*) was constructed to explain the several design factors above. One way to empirically capture these behaviors in a *FoM* was by using the following steps:Construct a nonlinear equation that fits the relationship between the trench and the sensitivityDivide the Equation by the beam thicknessMultiply the Equation by the gap distance

Following these steps, the Equation is represented by:(3)$$FoM=\frac{gap\times \left(-0.002\times {\left(starting\;position\right)}^{2}+0.0641\times \left(starting\;position\right)-0.5404\right)}{Thickness}$$

Defining $ratio=\frac{gap}{Thickness}$, the Equation is reduced to:(4)$$FoM=ratio\times \left(-0.002\times {\left(starting\;position\right)}^{2}+0.0641\times \left(starting\;position\right)-0.5404\right)$$
and can be plotted in a 3D plot, as shown in Figure 12.

The thickness and the position of the SiO_2_ layer were also investigated. It was found that increasing the SiO_2_ layer thickness linearly increased the sensor sensitivity. More importantly, placing the SiO_2_ layer on top of the beam allowed for an extensive range of motion and more sensitivity than placing it underneath.

The presented method for detecting helium can be very sensitive. As shown in Table 3, 0.05% of helium could produce a 0.15 µm displacement change in the beam deflection. Theoretically, using the holographic MEMS analyzer [24] with a 5 pm resolution, a helium change up to (0.05%/(0.15 µm/5 pm)) 2 ppm could be detected using the method presented in this work. This result is summarized in Table 4 and is compared with other methods that used thermal conduction to detect helium in the literature.

Finally, the presented approach provides computational proof of concept for a capacitive gas sensor. Future work will combine the thermal conductivity concept with our new concept of using electrical resonance activation for helium sensing [26].

## 5. Conclusions

This work provides a finite element model using COMSOL Multiphysics software to design and model a bimorph MEMS sensor for detecting helium in ambient air based on the thermal conductivity difference between the two gases. The thermal conductivity of helium is several times larger than that of air, so a device surrounded by helium will have reduced temperature and less movement. The sensor is designed from a poly-silicon layer as the base beam and an SiO_2_ layer with a thickness of 0.5 µm. Different design parameters are explored, and the best design is found when the MEMS beam’s thickness is 2 µm, the trench silicon starts at 160 µm, and the SiO_2_ layer is placed above the poly-silicon. The design achieves more than 4.28 µm sensitivity at the full helium conditions.

## Figures and Tables

**Figure 1 sensors-24-03626-f001:**
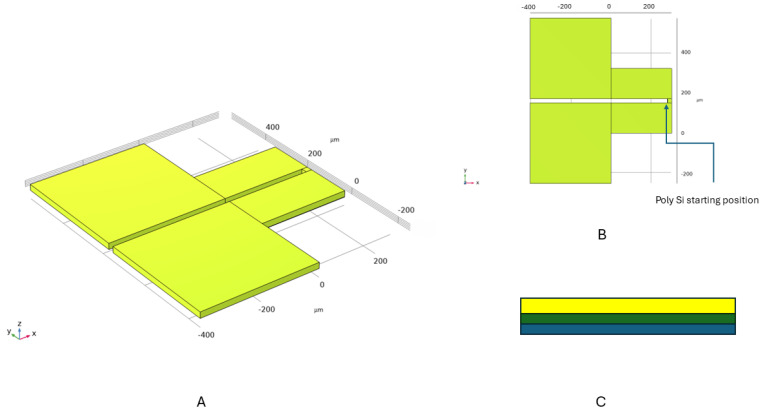
The geometry design of the bimorph MEMS device. (**A**) 3D geometry of the MEMS sensor. (**B**) Top view of the sensor. (**C**) Exaggerated side view of the sensor; yellow: poly-silicon; green: SiO_2_ layer; blue: sensor to substrate gas gap.

**Figure 2 sensors-24-03626-f002:**
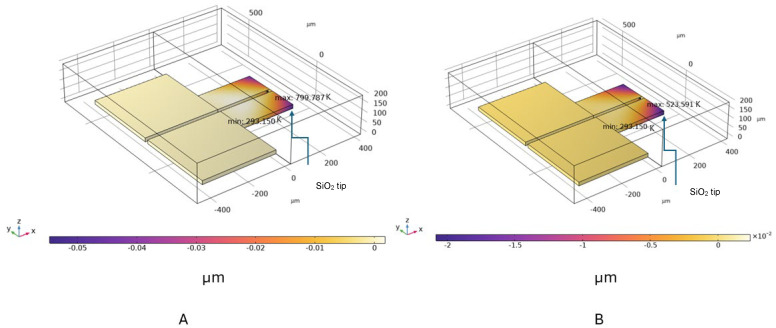
Displacement distribution for the MEMS sensor in the *z* direction: (**A**) air condition and (**B**) helium condition.

**Figure 3 sensors-24-03626-f003:**
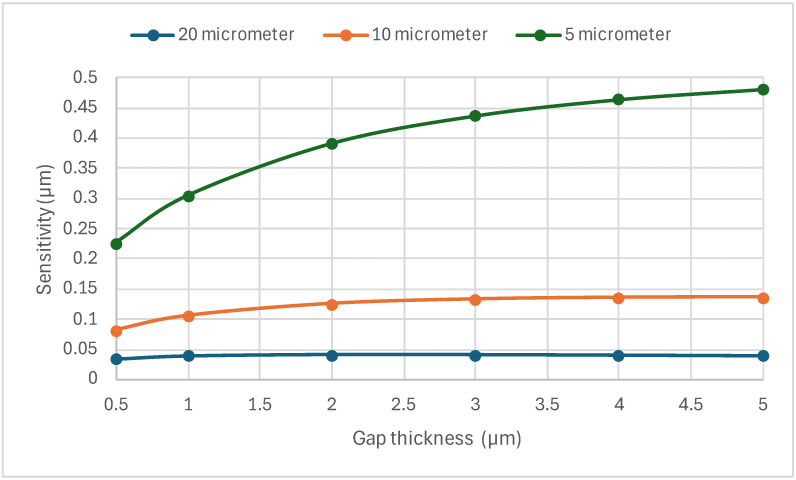
The effect of gap and beam thickness on the sensitivity of the MEMS sensor.

**Figure 4 sensors-24-03626-f004:**
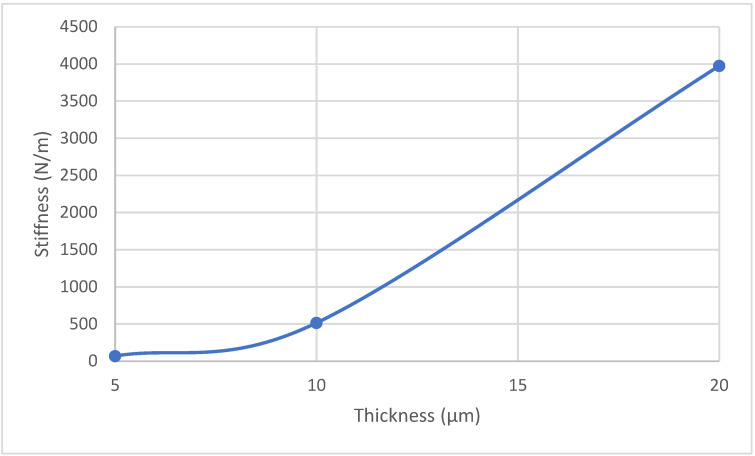
The stiffness of the MEMS sensor based on poly-silicon material properties and the thickness of the beam as a cantilever configuration. The stiffness is calculated by subjecting the sensor to a load, and then the displacement in the *z*-axis is recorded.

**Figure 5 sensors-24-03626-f005:**
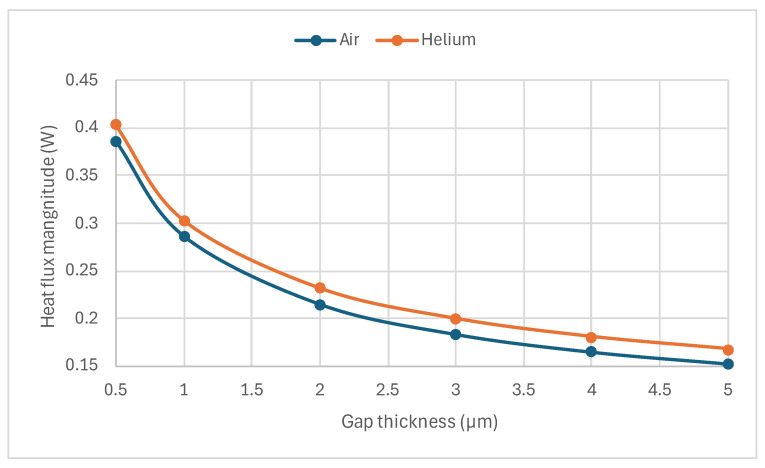
Heat flux magnitude for air and helium for the 5 µm poly-silicon beam thickness as a function of gap thickness.

**Figure 6 sensors-24-03626-f006:**
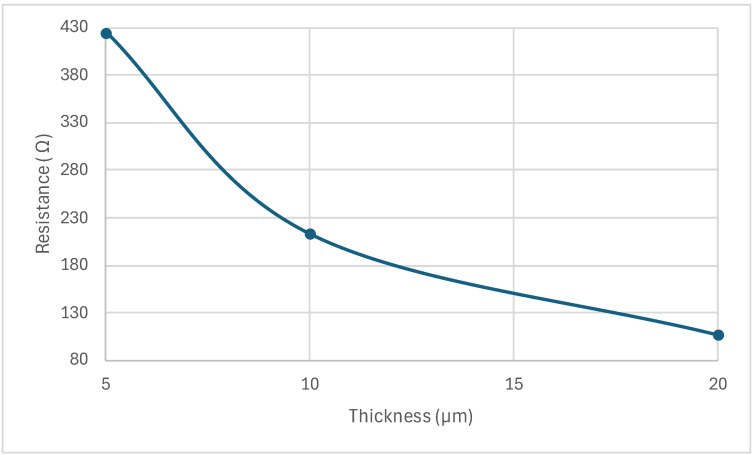
The electrical resistance of the MEMS sensor was modeled with poly-silicon material with a resistivity of 0.03 Ω cm.

**Figure 7 sensors-24-03626-f007:**
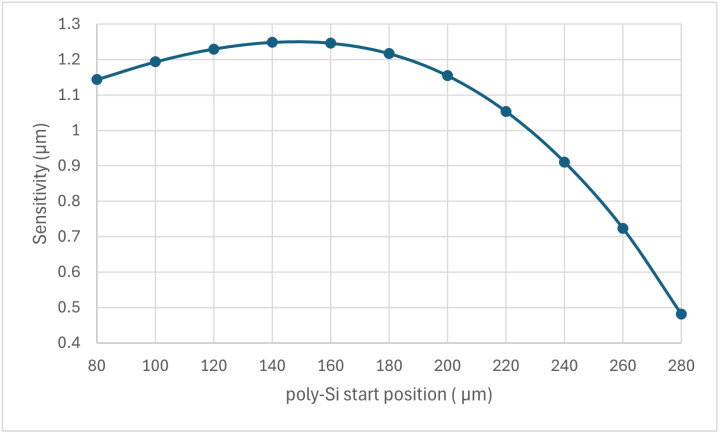
The sensitivity of the bimorph MEMS sensor as a function of the trench configuration.

**Figure 8 sensors-24-03626-f008:**
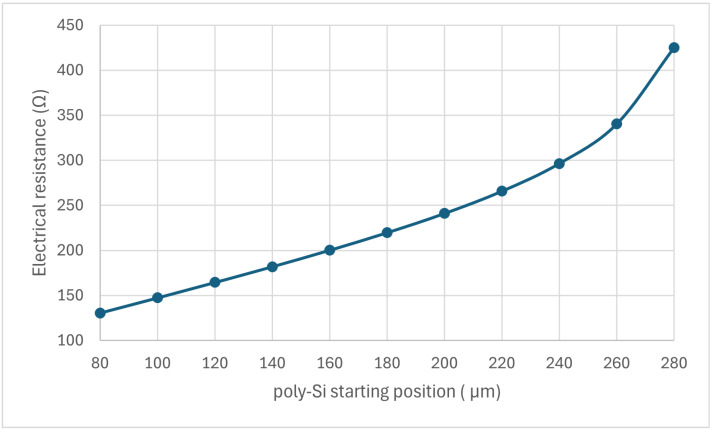
The electrical resistance of the MEMS sensor based on the trench configuration.

**Figure 9 sensors-24-03626-f009:**
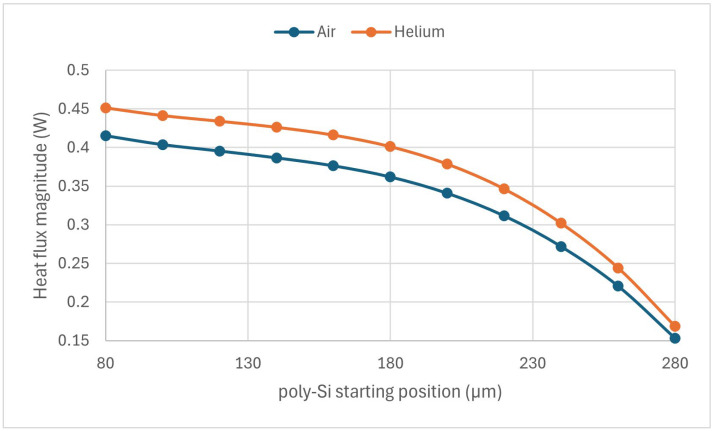
The magnitude of the heat flux for both air and helium based on the trench configuration.

**Figure 10 sensors-24-03626-f010:**
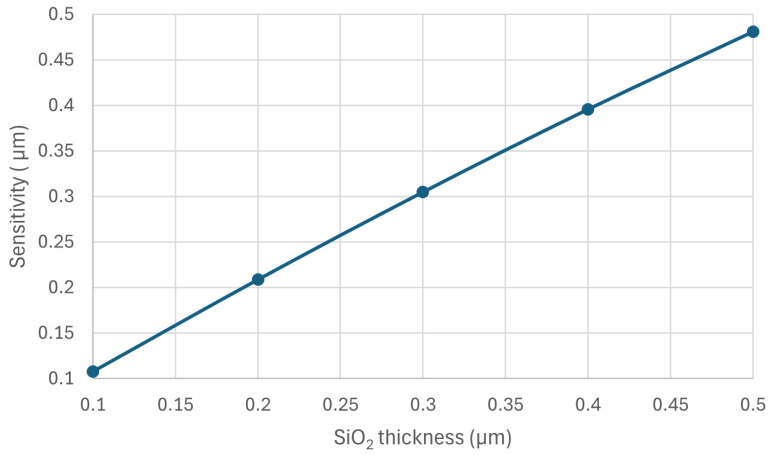
The sensitivity of the 5 µm poly-silicon beam as a function of the SiO_2_ layer thickness.

**Figure 11 sensors-24-03626-f011:**
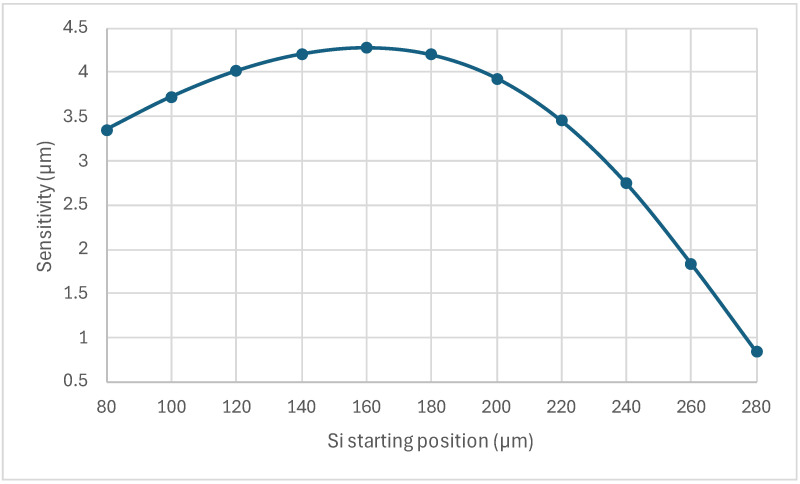
Sensitivity of the 2 µm poly-silicon beam with a 0.5 µm SiO_2_ layer above it as a function of the trench configuration.

**Figure 12 sensors-24-03626-f012:**
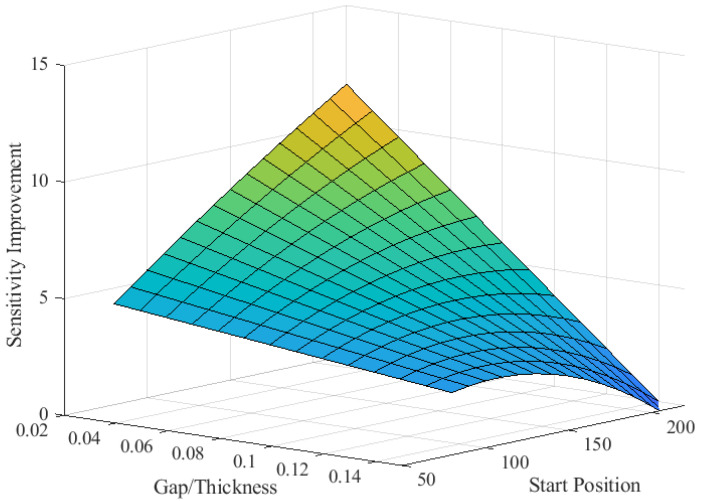
Plot of the figure of merit (*FoM*) showing the effect of the design variables (start position, beam and gap thickness) on the sensitivity of the sensor.

**Table 1 sensors-24-03626-t001:** The input voltage required to reach the 800 K limit for each design.

Beam Thickness (µm)	Gap Thickness (µm)	Input Voltage (V)
5	0.5	13.234
5	1.0	11.504
5	2.0	10.105
5	3.0	9.418
5	4.0	8.989
5	5.0	8.688
10	0.5	11.365
10	1.0	9.991
10	2.0	8.875
10	3.0	8.347
10	4.0	8.027
10	5.0	7.809
20	0.5	9.896
20	1.0	8.807
20	2.0	7.96
20	3.0	7.583
20	4.0	7.365
20	5.0	7.221

**Table 2 sensors-24-03626-t002:** Input voltage for the MEMS sensor based on the trench configuration.

Si Start Position (µm)	Input Voltage (V)
80	8.16
100	8.537
120	8.904
140	9.248
160	9.553
180	9.975
200	9.941
220	9.953
240	9.789
260	9.399
280	8.688

**Table 3 sensors-24-03626-t003:** Displacement of the MEMS sensor based on the 2 µm thick poly-silicon beam and 160 µm silicon starting position.

Helium Volume Fraction	Displacement (µm)	Displacement Difference (µm)
0	5.30	0
0.01	5.11	0.19
0.02	4.92	0.19
0.03	4.75	0.17
0.04	4.59	0.16
0.05	4.44	0.15

**Table 4 sensors-24-03626-t004:** Comparison of some thermal conductivity helium sensors based on the minimum detection level.

Ref.	Minimum Detection Level
[20]	0.07%
[25]	5%
This work	Up to 2 ppm (2 × 10^−6^%)

## Data Availability

Data are contained within the article.

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
