# Peer review of "Modeling and Design Parameter Optimization to Improve the Sensitivity of a Bimorph Polysilicon-Based MEMS Sensor for Helium Detection"

_sensors, 2024, doi:10.3390/s24113626_

Round 1

Reviewer 1 Report

Comments and Suggestions for Authors

In this paper, the authors designed and modeled a bimorph MEMS sensor for detecting Helium in ambient air using COMSOL Multiphysics software, and achieved a high sensitivity by optimizing design parameters. The study provides a methodology for the research of a novel MEMS gas sensor using the thermal conductivity difference, the paper is logical with detailed data and analysis, but the paper needs some improvements before accepting it for publication, and my detailed comments are as follows:

1. Please give the version of COMSOL simulations software.

2. Page 2, line 70, please add the unit of the pad size.

3. Please supplement the working principle formula of the MEMS sensor.

4. Please represent and label the SiO2 layer in Figure 1 and Figure 2.

5. Page 3, paragraph 3, in the last sentence, whether the sensitivity and unit setting in the text are appropriate.

6. Please supplement the electrical output scheme of the sensor.

7. Please change the appropriate horizontal coordinate name in Figure 7, Figure 8 and Figure 9.

8. Page 9, paragraph 2, in the last sentence, please modify the correct units for thickness.

Comments on the Quality of English Language

Minor editing of English language required.

Author Response

We value the reviewers comments, and we tried our best to address 

Reviewer 2 Report

Comments and Suggestions for Authors

Comment 1: The authors claim that the proposed thermal conductivity sensor will be able to detect He by comparing the thermal conductivity of this inert gas with Air. However, I believe, you cannot neglect thermal convection in air. Can the authors quantify both mechanisms for the sensor geometry of study?

Comment 2: Can the authors compared their technology with the state-of-the-art (SoA) thermal conductivity sensors? See examples below.

https://us.msasafety.com/p/DTX832-HE?locale=en

https://www.mtl-inst.com/product/k522_-_thermal_conductivity_tcd_gas_sensor_for_hydrogen_h2_helium_he_argona/

Comment 3: The Introduction describes how the thermal conductivity sensor bends as a function of the surrounding gas thermal conductivity. However, it does not say what electrical signal you intend to measure (capacitance variation, tip displacement using a laser beam, etc.)?

Comment 4: The authors should highlight on Fig. 1 (e.g., using different colors) every structural component forming part of the sensor. For example, Si and SiO2 layers, cantilever beam, trench, substrate-to-cantilever gap, etc.

Comment 5: The authors state that “the MEMS device was subjected to DC heating voltage to obtain a maximum temperature of around 800 K in the air condition to minimize the creep effect”. Can the authors also specify the temperature threshold at which you expect poly-Si start creeping?

Comment 6: Fig. 2 shows the displacement of the cantilever beam. However, I was expecting that the entire protrusion would flap. Instead I only see the edges bending down while the center stays horizontal. Is the center made of different materials? I believe my question would be answered by having a clearer structural diagram of the sensor (see comment 4).

Comment 7: Fig. 3 reports the sensitivity of sensor vs. gap thickness. However, in a sensor, the output magnitude should be electrical. Can the authors replace displacement by the electrical magnitude expected at the output? This relates to Comment 3 of measured signal.

Comment 8: How is the stiffness calculated in Fig. 4? Is this magnitude relevant for sensitivity?

Comment 9: It is not clear to me the paragraph prior to Fig. 5. My understanding is that you want to achieve higher heat flux so your sensor is more sensitive to the presence of gas. However, the authors state here that “sensitivity increases as the gap distance increases”. But if the gap distance increases too much, thermal convection will start dominating. Can you please clarify? Perhaps, this can be answered by quantifying both phenomena at the beginning of discussion.

Comment 10: In the Discussion section, the authors state that thinner beams help to lower stiffness, but increase the resistance, so the input voltage. Since several design factors are playing a role, I would define a Figure of Merit (FoM) that encompasses all of them with a positive slope. The FoM can be used to compare among technologies.

Comment 11: At the end of the paper, I would include a table comparing this design with other thermal capacitive sensor technologies.

Comment 12: If the sensor is better than the SoA, shall it be fabricated and proved experimentally?

Author Response

We value the reviewer's comments and we tried our best to address 

Round 2

Reviewer 2 Report

Comments and Suggestions for Authors

Thank you for addressing all my comments. In my opinion, the paper can be published in the present form. Good job!